# Factors explaining resilience among nepalese nurses of tertiary-level hospital experiencing COVID-19 pandemic: A cross-sectional study

Sarala KC[1], Rekha Timalsina [2,3]*, Shanta Dangol Shrestha[4], Praneed Songwathana[5]

**1** Co-Leads Registered Nurse (RN), Master in Nursing (MN), Professor, Former Dean, Patan Academy of Health Sciences (PAHS), School of Nursing and Midwifery [SONAM], Sanepa, Lalitpur, Nepal, **2** Co-Leads RN, MN, MA, PhD in Nursing Science, Associate Professor, PAHS, SONAM, Sanepa, Lalitpur, Nepal, **3** National Institute on Ageing and Health (NIAH), Lalitpur, Nepal, **4** RN, MN, Nursing Director, PAHS, Patan Hospital, Sanepa, Lalitpur, Nepal, **5** RN, PhD, Professor, Faculty of Nursing, Prince of Songkla University, Hat Yai, Thailand

* rekhatimalsina@pahs.edu.np; rekha.timalsina@gmail.com

## Abstract

Resilience is important for nurses in preventing mental health problems, promoting psychological well-being, enhancing the quality of patient care, and empowering them to effectively manage healthcare crises. Fostering resilience and identifying its explanatory factors is essential for the nursing profession while facing persistent challenges during and after the COVID-19 pandemic. Considering the scarce evidence, this study aimed to identify the current state of resilience and its explanatory factors among Nepalese nurses at a tertiary hospital during the COVID-19 pandemic. Cross-sectional research was conducted among 307 nurses (i.e., staff nurses and sisters in-charge) from tertiary hospital 'A', selected using proportionate stratified random sampling. The data were collected via a self-administered structured questionnaire using a socio-demographic and job-related information sheet and five other sets of standard, valid, and reliable instruments. The data were analyzed using descriptive statistics and inferential statistics, namely Pearson Product-Moment Correlation with multiple linear regression and path analysis. The highest percentage of respondents (51.8%) had an intermediate [i.e., neither low nor high] level of resilience. The model of resilience fits with the empirical data. Furthermore, self-efficacy, perceived social support (PSS), and compassion satisfaction (CS) were the statistically significant factors that explained 31 percent of the variance in resilience (*Adjusted $R^2$* =.31) with other non-significant factors (i.e., perceived organizational support and burnout). Additionally, PSS and CS had statistically significant positive indirect effects on resilience through self-efficacy, with their total effects. In conclusion, nurses had an intermediate level of resilience, and the resilience model aligned well with the empirical data. Therefore, hospital and nursing administration should consider these findings to design and implement targeted interventions that foster resilience. This can

**Data availability statement:** All relevant data are within the paper and its Supporting information files.

**Funding:** This work was supported by the Patan Academy of Health Sciences [PAHS], for the grant under the Prof. Dr. Neelam Adhikari Memorial Fund [01/8th July 2022]. This project was co-leads by Prof. Sarala K. C. and Dr. Rekha Timalsina under this grant. However, the funders had no role in study design, data collection and analysis, decision to publish, or preparation of the manuscript.

**Competing interests:** The authors have declared that no competing interests exist.

be achieved by preventing burnout and strengthening nurses' positive psychosocial resources, including compassion satisfaction, self-efficacy, and social and organizational support, to help them navigate the challenges of their demanding profession.

## Background

The mental health, well-being, and work effectiveness of healthcare workers (HCWs) [1] were influenced by the Coronavirus Infectious Diseases (COVID-19) pandemic [2]. In particular, HCWs experienced heightened uncertainty regarding disease exposure, stigmatization, and the potential risk of transmitting COVID-19 to their families [3]. Likewise, nurses, as frontline workers, faced different psychological [4–7], social [5–7], emotional [5,6], and physical [7] challenges while taking care of patients during the pandemic. As a result, COVID-19 particularly profound effect on the mental health of frontline workers [1,8], as well as their interpersonal relationships with colleagues, family, and society at large [8]. Moreover, HCWs and social care professionals faced a variety of stressors while delivering care for patients [8]. Several factors contributed to these problems, including limited resources [4,5]; fear of infection [4]; rejection by acquaintances and friends [5]; increased work demands, diminished social relationships, and changes in routine life [6]. In addition, the pandemic significantly altered the healthcare work environment and escalated job-related demands [3]. In the Nepalese context, the first COVID-19 case was detected on 23 January 2020. Since then, Nepalese nurses have actively engaged in caring for patients throughout the first, second, and third waves of the COVID-19 pandemic [9]. Consequently, the Nepalese people have continuously experienced the impacts of the pandemic [10].

Similarly, Nepalese frontline HCWs, including nurses, encountered numerous challenges during this period, such as a lack of coordination between public and private facilities [11], operational constraints at local and provincial levels [11], insufficient resources [9,11], and a lack of incentives or safety measures [11]. Furthermore, HCWs were stigmatized by house owners, neighbors, police, the public, and the media [12] due to their perceived risk of spreading the virus [12,13]. As a result, these experiences led to various psychological issues such as anxiety, depression, and related conditions [14–19], work-engagement-related issues [19], psychological distress, and moral injury among HCWs [12] in the international and Nepalese contexts. Nevertheless, despite these challenges, HCWs remained motivated by professional ethics, peer and family support, and a moral obligation to deliver quality and dignified care to patients [11].

Given these circumstances, Virga and Palos [20] emphasized the importance of studying positive psychological concepts during the pandemic. Among such constructs, resilience stands out as a vital attribute. It reflects an individual's capacity to cope effectively with the adverse effects of stress, thereby preventing mental health problems [8,21], and fostering mental health [1,2,8], enhancing adaptation to challenging situations, and improving overall quality of work life [22]. For instance, a cross-sectional study in India during the COVID-19 pandemic revealed that 47.5%

of nurses had a high level of resilience [22]. Importantly, resilience is a multidimensional construct [20] that may be influenced by both external and internal stressors [23], such as the working environment [23,24], different psychological stressors [20], organizational dynamics [23], and individual personal attributes [23].

In this regard, systematic reviews, cross-sectional, and qualitative studies have reported a statistically significant negative relationship between various occupational stressors and resilience among nurses. These include burnout [25–30]; stress [25,27]; secondary traumatic stress [28,29]; post-traumatic stress disorder, workplace bullying [27]; and compassion fatigue [31]. In contrast, several factors have demonstrated a significant positive relationship with resilience among nurses and healthcare workers across different countries. These include compassion satisfaction [28–30]; job designation and perception of global transformational leadership [32]; self-efficacy [27,33]; perceived organizational support (POS) [34]; coping skills, job retention, social support, general well-being [27]; and job satisfaction [25,27]. Moreover, managerial innovation, proactive behavior, and locus of control of nurses directly and indirectly influence resilience [24]. Notably, compassion satisfaction had a statistically significant relationship with self-efficacy [35]. Likewise, Hobfoll [36] stated the indirect positive relationship of social support with resilience via self-efficacy.

In addition, resource support is another critical factor influencing resilience among nurses, as highlighted in an integrative review [37]. However, it is important to note that low- and middle-income countries (LMICs) experienced even greater resource constraints than high-income countries [38], including Nepal [39–41]. Despite extensive research on healthcare leadership, management practices, work environments, and the adverse psychological effects of the COVID-19 pandemic, a notable gap remains. There is limited understanding of the long-term psychosocial experiences of nurses, regarding positive psychological outcomes, such as their readiness to manage future infectious disease disasters, mental well-being, and resilience. However, in LMIC settings like Nepal, the relationship of compassion satisfaction, burnout, secondary traumatic stress, perceived organizational support, social support, and self-efficacy, with resilience among nurses remains largely underexplored. This gap is reflected by Xu et al. [6], who highlighted the need for further studies on this issue, and Pollock et al. [8], who emphasized the importance of well-conducted research to improve mental health of frontline workers. Thus, the present study was designed to identify the factors explaining resilience among nurses working at a tertiary-level hospital, Nepal.

### Conceptual framework of this study

The research team selected the relevant factors and developed a conceptual framework to support the hypothesized model of resilience in this study based on the empirical investigation and findings of the systematic reviews, including the work of Timalsina et al. [42]. To ensure rigor, the researcher critically analyzed the relevance and significance of resilience-associated factors among Nepalese nurses during COVID-19, avoiding a kitchen-sink approach and minimizing Type I errors, as suggested in the literature [43]. Ultimately, this study focuses on assessing positive psychological concepts, particularly resilience and its explanatory factors, among Nepalese nurses in a tertiary hospital setting. Consequently, these findings will raise awareness among policymakers, nurses, and healthcare authorities to develop interventions that enhance self-efficacy, organizational support, and resilience while mitigating burnout and secondary traumatic stress. Thus, the study contributes to ensuring high-quality patient care and long-term mental well-being among nurses.

## Methods

### Aim and objectives

This study aimed to identify the factors explaining resilience among Nepalese nurses in a tertiary-level hospital during the COVID-19 pandemic. Likewise, the objectives of this study were to; (1) identify the level of resilience, (2) examine the goodness-of-fit of the hypothesized model of resilience (Fig 1) with the empirical data; and (3) analyze the direct and positive effects of compassion satisfaction, perceived social support, perceived organizational support, and self-efficacy on

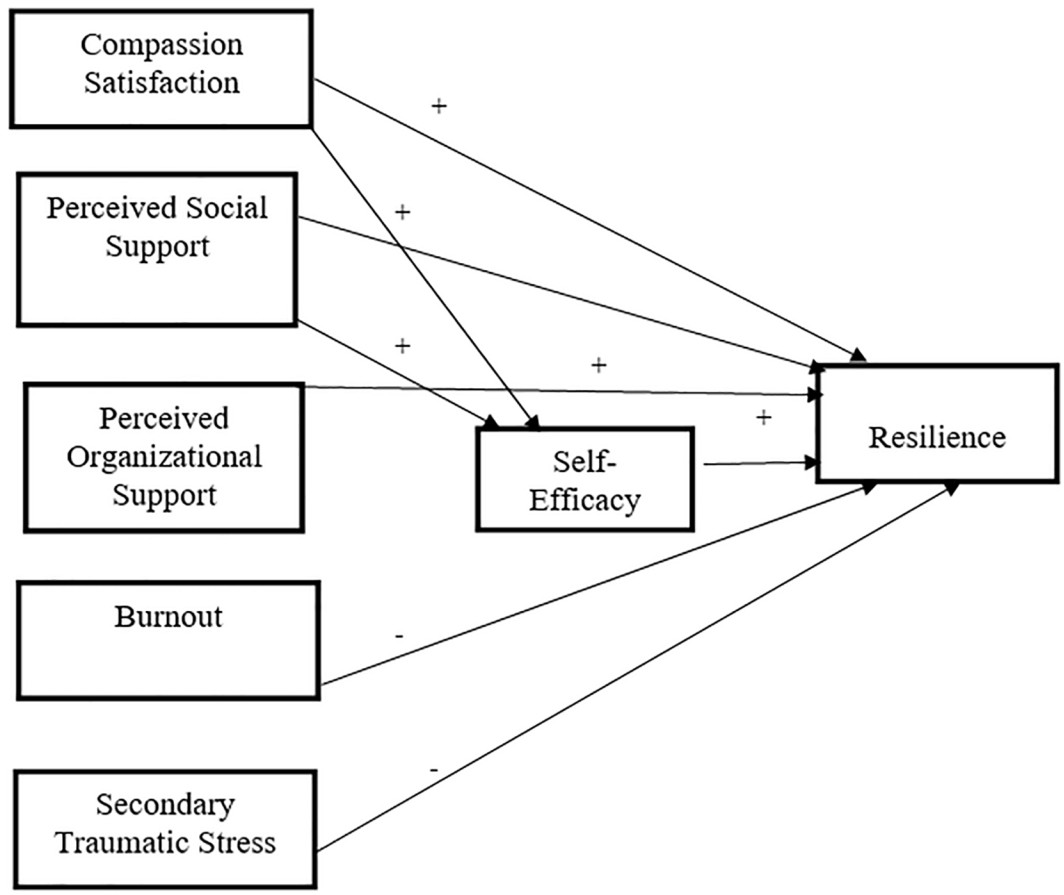

**Fig 1. Conceptual framework on hypothesized model of resilience among Nepalese nurses.**

resilience; (4) examine the direct and negative effects of burnout and secondary traumatic stress, on resilience, (5) assess the indirect and positive effects of compassion satisfaction and perceived social support on resilience through self-efficacy among nurses of working in a tertiary hospital in Nepal experiencing the COVID-19 pandemic (Fig 1).

## Study design

Due to the unavailability of data on pre-pandemic, early-pandemic, and post-pandemic resilience among nurses, the research team was unable to track dynamic changes in resilience over time. Therefore, a cross-sectional path analytical study was conducted in adherence to Strengthening the Reporting of Observational Studies in Epidemiology [STROBE] guidelines.

## Study setting

The study was conducted at Tertiary Hospital 'A' in Lalitpur, one of the largest and major teaching hospitals in Nepal. This hospital was designated for the health care services for the people from every district in remote and urban settings of Nepal. Additionally, this hospital took a pivotal role in providing care to patients with COVID-19 and non-COVID patients simultaneously from the initial phase of the pandemic to the fourth wave of COVID-19.

## Study population

The study population of Tertiary Hospital 'A' was 441 nurses (i.e., 413 staff nurses and 28 sisters in-charge), excluding nursing administrators, supervisors, and nurses on extended study or maternity leave, along with 37 newly recruited staff nurses. The nurses from various departments, who were employed full-time, actively worked in the hospital during the COVID-19 outbreak, and had a minimum of six months of work experience, were included in this study. However, nurses who declined to participate (n = 23) in the study were excluded.

**Sample Size Calculation. (A) The initial sample size calculation** was performed using the formula cited by Berman [44]: no = {[($z^2pq$) + ($ME^2$)]/ [$ME^2$ + ($z^2pq$/ N)]}, where, Z = 1.96 for 95% confidence level, p = 47.5% [high level of resilience based on evidence by Jose et al. [22], q = 1-p (i.e., 52.5%), ME (Margin of error) = 5%, no = Sample Size, and N = Population Size (i.e., 441). Thus, the calculated sample (no) was 206 nurses. **(B) Adjusting for non-response (n'),** the following formula was used: n' = no + no of 28.2%: n' = 206 + {[206 × 28.2%] (i.e., 58.1)} (based on 28.2% of non-responses in the questionnaire method of data collection as per the prior evidence [45]. Thus, the sample size (n') = 264 nurses. **(C) Sample Size After Adjusting for Power (n\*).** Finally, the adjusted sample size was calculated by the following formula: n\* = n'/0.8 (i.e., n\* = 264/0.8) as applied in the prior study [46]. This adjustment, reflecting an 80% power, was aimed to increase the sample size to reduce the Type II error and maintain a balance between Type I and Type II errors, thereby reducing the likelihood of one error type without increasing the likelihood of the other. Therefore, the final sample size was 330 nurses.

## Sampling

Firstly, with the help of the authority of the nursing administration of tertiary hospital 'A', the research team prepared the sampling frame. The lists consist of all the nurses working at Tertiary Hospital A, who met the eligibility criteria with their years of experience and job designation. Secondly, the researchers employed a proportionate stratified random sampling to select a precise and representative sample of nurses based on two distinct groups (i.e., strata): staff nurses and sisters in-charge, and to ensure the possibility of generalizing the findings in the selected study setting. Thus, the strata sample size was calculated using the formula: nh = (Nh/ N) × n [47], where nh represents the sample size for stratum h, Nh stands for the population size for stratum h, N denotes the total population size, and n signifies the total sample size. Hence, 330 nurses were selected [309 staff nurses and 21 sisters in-charge]. Thirdly, samples were selected from each stratum (i.e., staff nurses and sisters-in-charge) using a simple random sampling technique, employing random numbers generated by a random number generator. Despite implementing strategies to minimize non-responses such as follow-up, direct contact, and providing flexible timelines for questionnaire returns, 23 nurses declined participation due to personal reasons. Consequently, the data were collected from 307 nurses, rather than the initially targeted sample size of 330, resulting in a dropout rate of 6.97% (Fig 2).

## Instrumentation

The instruments used in this study are described as follows:

**Socio-demographic and job-related information.** The socio-demographic and job-related information encompassed eight structured items with open-ended and closed-ended formats. Open-ended questions were about age, years of experience within the current organization, and the current working department. Conversely, closed-ended questions were employed for marital status, educational level, current designation, and type of appointment, each structured with binary Yes/No options.

**Connor-davidson resilience scale-10 (CD-RISC-10) [48].** In 2003, Kathryn M. Connor and Jonathan R. T. Davidson developed the initial 25-item Connor-Davidson Resilience Scale (CD-RISC) [49], and subsequently, Campbell-Sills and Stein [48] reexamined its factor structure in undergraduate students, suggesting a condensed 10-item version of

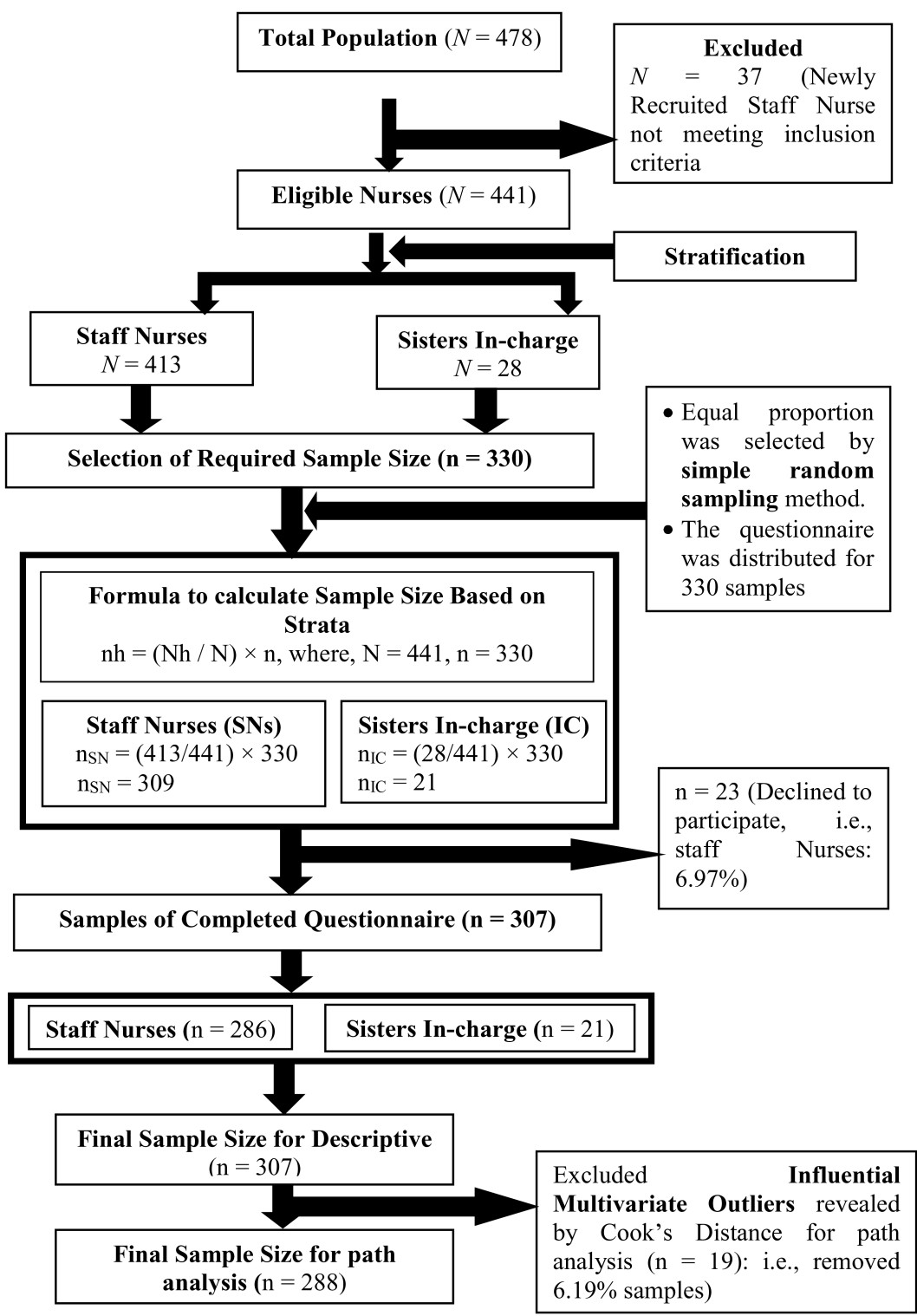

**Fig 2. Flow diagram of sample recruitment, selection, and final sample size.**

CD-RISC. Furthermore, CD-RISC-10 was validated in the Nepalese context and named as CD-RISC-10-N [50]. Thus, this study used a CD-RISC-10-N to assess the resilience of nurses, evaluating their perceived capacity to maintain physical, psycho-social, and spiritual equilibrium amidst the challenges. One of the examples of items from the scale is *"I can deal with whatever comes my way."* The scale comprises 10 items, each rated on a five-point Likert scale ranging from Not True at All (0) to True Nearly All the Time (4). The total scores range from 0 to 40. Resilience was classified into three levels: low (up to the first quartile), intermediate (the second and third quartiles), and high (fourth quartile) [51]. CD-RISC-10 was found to be reliable among nurses during the COVID-19 pandemic [29,52].

**General Self-Efficacy Scale (GSE-10) [53].** Ralf Schwarzer and Matthias Jerusalem developed the English version GSE-10 in 1995 to measure optimistic self-beliefs that help people cope with life's challenges [53,54]. GSE-10 was employed to evaluate nurses' self-efficacy, encompassing their confidence and belief in their ability to think critically, confront and solve problems, achieve goals, stay calm in stressful situations, and effectively manage challenges. An illustrative item from the scale is, *"When I am confronted with a problem, I can usually find several solutions."* The GSE-10 uses a four-point rating scale, ranging from All True (1) to Exactly True (4), with total scores ranging from 10 to 40. Self-efficacy was categorized as low (below the median score), medium (at the level of the median score), and high (greater than the median score) [54]. The GSE-10 was valid and reliable [53] and was also reliable among nurses [55] and other healthcare professionals [56] experiencing the COVID-19 pandemic, including the Nepalese older adults [42].

**Multi-dimensional Scale of Perceived Social Support (MSPSS-12) [57].** Zimet et al. [57] initially developed and demonstrated factorial and construct validity of MSPSS with 12 items, 7-point Likert format, ranging from Very Strongly Disagree (1) to Very Strongly Agree (7). MSPSS-12 consists of three subdomains: (1) significant others (i.e., neighbors, healthcare providers, or any person excluding family members or friends), (2) family (i.e., parents, grandparents, children, or siblings), and (3) friends. Additionally, the MSPSS was translated into the Nepali language and found to be reliable and valid among Nepalese people and named as MSPSS-N [58]. Thus, 12-item MSPSS-N was used to measure perceived social support among nurses. An illustrative item from the scale is, *"I get the emotional help and support I need from my family."* The score ranges from 12 to 84. Perceived social support was divided into low (<Mean -1SD), Moderate (between Mean±1SD), and High (> Mean + 1SD) [59]. The MSPSS demonstrated strong reliability among Chinese nurses facing the COVID-19 pandemic [60].

**The Survey of Perceived Organizational Support (SPOS) [61].** *SPOS* was developed in 1986 by Eisenberger et al. [61], and the widely used 8-item version is a shortened, validated form of the original 36-item scale. SPOS was used to evaluate nurses' perceived support from their employer organization. SPOS measures nurses' perceptions of how much their organization values their contributions and cares about their well-being. Nurses rated their experiences over the preceding 30 days on a scale from Strongly Disagree (0) to Strongly Agree (6). The scale includes four negatively worded items, which were reverse-scored to compute the aggregate SPOS score. An example of items is, *"This hospital administration values my contribution to its well-being."* The score ranges from 12 to 84. Perceived organizational support was classified into low (1 standard deviation below the mean), moderate (between ± 1 standard deviation), and high (1 standard deviation above the mean) [62]. Worley et al. [63] showed that the SPOS 8-item scale was valid and reliable. Additionally, other prior studies demonstrated the reliability of SPOS among nurses in the USA [64] and China [65], as well as among Nepalese nurses [66] and nursing faculty [46].

**The Professional Quality of Life Scale Version 5 (ProQoL 5) [67].** The Professional Quality of Life Scale was initially developed by Stamm in the late 1990s with the current ProQoL version 5 released in 2010 [67] and translated into Nepali language by Adhikari [68] (i.e., ProQoL-5 Nepali version). The ProQoL has established validity and reliability during its developmental stage as cited in Stamm [67]. ProQoL-5 aims to assess professionals exposed to traumatic situations, including nurses, doctors, psychologists, teachers, and social workers. ProQoL-5 comprises three sub-concepts, i.e., compassion satisfaction, secondary traumatic stress, and burnout, each containing 10 items, designed to capture distinct aspects of their professional well-being. Thus, ProQoL-5 was used to measure the positive and negative experiences

of the nurses in the last 30 days during the COVID-19 pandemic. The overall score of ProQoL-5 ranges from 50 to 150. Thus, ProQoL-5 was used to measure the positive and negative experiences of the nurses in the last 30 days during the COVID-19 pandemic. **Compassion satisfaction [CS]**, for instance, evaluates the positive emotions associated with job satisfaction, such as happiness, success, and the desire to make a difference while adapting to new technologies and protocols. An example item is, *"I get satisfaction from being able to help people."* Conversely, **Secondary Traumatic Stress [STS]** addresses the negative emotions stemming from feeling overwhelmed and fearful due to the demands of caring for others. An example item is, *"I am preoccupied with more than one person I help."*

Finally, **Burnout [BO]** includes equally five negatively and positively worded items that capture feelings of unhappiness, detachment, and insensitivity towards the work environment. Consequently, this sub-concept reflects exhaustion and a sense of being overwhelmed. An example item is, *"I am not as productive at work because I am losing sleep over the traumatic experiences of a person I help."* The response options of ProQoL-5 range from Never (1) to Very Often (5), with each sub-concept score ranging from 10 to 50. Raw Total Scores of CS, BO, & STS were converted into z score and then converted into t-scores using these {i.e., $tCS = [(zCS \times 10) + 50]$; $tBO = [(zBO \times 10) + 50]$; and $tSTS = [(zSTS \times 10) + 50]$} equations [67]. Compassion satisfaction, secondary traumatic stress, and burnout were categorized as Low = Below the 25th Percentile, Moderate = 25th to 75th Percentile, and High = Above the 75th Percentile [67]. Furthermore, prior studies among healthcare professionals [29] and nurses [69,70] demonstrated the reliability of ProQoL, including its overall scale and subscales. Additionally, ProQoL 5-N has been successfully applied among Nepalese Mental Health and Psychosocial Support professionals [68], further supporting its versatility and robustness across various professional contexts.

Three experts: (1) a nursing administrator, (2) a nurse academician/ college administrator, and (3) a nurse supervisor assessed the adequacy of the content and relevancy of the items of the CDRISC-10, MSPSS, GSE-10, SPOS, and ProQoL-5, considering the Nepalese culture context. The overall average scale content validity index (S-CVI/Ave) of all these scales was greater than .90 based on the experts' ratings, which reflected the excellent content validity of these instruments. The instruments were translated into the Nepali language through a rigorous forward and backward translation process by prior researchers (i.e., MSPSS-N [58], SPOS [66], ProQoL-N [68], CDRISC-10 [50], and GSE-10 [42]). However, two nursing graduates in the current study initially reviewed both the English and Nepali versions of the instruments to ensure linguistic clarity and simplicity.

Additionally, these translated versions of CDRISC-10, MSPSS-N, and GSE-10 were used among Nepalese Older Adults Disaster survivors and found to be reliable [42]. Subsequently, pretesting of the Nepalese version of instruments was carried out among 41 nurses at Tertiary Hospital 'B' of Kathmandu, chosen for their similarity in characteristics to the intended sample for this study. The Nepali versions of the CDRISC-10-N, MSPSS-N, GSE-N, SPOS-N, and overall ProQoL-N signified robust internal consistency reliability based on data obtained in the pretesting. The Cronbach's alpha values obtained from the pretesting data of 41 nurses are as follows: CD-RISC ($\alpha = .81$), GSE-10 ($\alpha = .89$), MSPSS ($\alpha = .96$), SPOS ($\alpha = .71$), and PRQoL [Overall scale that includes burnout, secondary traumatic stress, and compassion satisfaction] ($\alpha = .75$). Additionally, data from 307 samples in this study confirmed the internal consistency reliability of the instruments used, as follows: CD-RISC ($\alpha = .80$), GSE-10 ($\alpha = .84$), MSPSS ($\alpha = .94$), SPOS ($\alpha = .84$), and ProQoL ($\alpha = .75$). The subscales of ProQoL also demonstrated acceptable reliability: BO ($\alpha = .70$), STS ($\alpha = .79$), and CS ($\alpha = .84$).

## Data collection procedure

For approximately eight weeks, data were collected from the 16th of April to the 15th of June 2023. Before data collection, the research team conducted comprehensive discussions regarding the tools, administration methods, and instructions necessary for the respondents. Two research assistants (RAs), each holding a B.Sc. in Nursing and prior experience in data collection in hospital settings, were recruited. The research team rigorously trained RAs involving face-to-face interactions on guidelines, data collection procedures and instruments, checklists, and informed consent procedures. RAs received training in effective communication, emphasizing the importance of patience, clarity in asking questions,

and competence in collecting relevant information. Then, the RAs collected data through self-administered questionnaire methods (i.e., paper forms) by applying all the ethical procedures mentioned in the ethical consideration section.

Nurses, working across 30 different in-patient and out-patient departments, were the targeted respondents. The RAs visited at least two departments daily to meet respondents. One RA was scheduled to visit the department in the morning and another in the evening. RAs distributed questionnaires to each of the randomly selected respondents after obtaining informed consent from each of them. Respondents were encouraged to provide genuine responses without consulting colleagues. To facilitate efficient data collection and ensure maximum participation, the RAs provided extra time with flexible return times for those respondents who were unable to complete it during their clinical duties. Thus, the RAs followed up with phone calls at 3-day and 6-day intervals to collect the completed questionnaires from respondents who needed additional time to finish them. The RAs collected completed questionnaires directly from each respondent based on mutually agreed-upon dates and times. The average time for questionnaire completion was around 30 minutes. The RAs and the research team provided contact numbers to the respondents to provide an opportunity to ask any questions, resolve ambiguities, or replace misplaced questionnaires. This approach facilitated the collection of accurate data by clarifying uncertainties and minimizing missing data due to lost questionnaires. The RAs reviewed the questionnaires for completeness, correctness, and accuracy of responses in the presence of the respondents. The respondents were asked to respond with their consent to ensure data completeness in missing items. Furthermore, continuous communication between the research team and the RAs ensured supervision, guidance, and problem resolution throughout the data collection process.

## Data management and statistical analysis

Each questionnaire underwent thorough checks by RT for completeness and consistency after collecting all the questionnaires from the RAs. The data were meticulously edited, classified, and manually coded. Double data entry and cleaning were performed using Epi Data software. Following this, analysis was done in the Statistical Package of Social Sciences (SPSS) version 16 software (SPSS Inc., Chicago, III., USA) and Confirm IBM SPSS Analysis of Moment of Structures [AMOS] version 21 software (AMOS Development Corporation, USA). Univariate outliers of the observed factors of the instruments were managed by the winsorization method. Harman's Single Factor Test showed that a single component accounted for 17.88% of the variance across resilience, self-efficacy, perceived social support, perceived organizational support, compassion satisfaction, and burnout. This low variance suggests the data were free from common method bias associated with using a single data collection method (i.e., self-administered questionnaire with Likert scale items).

Descriptive statistics, including frequency, percentage, median, and interquartile range, mean and standard deviation, skewness, and kurtosis, were used to describe the sample characteristics and self-efficacy, compassion satisfaction, burnout, secondary traumatic stress, perceived social support, perceived organizational support, and resilience. Assumption tests were performed for Pearson Product-Moment Correlation analyses, ensuring measurement level compatibility, identifying related pairs, detecting outliers, and confirming linearity. Correlation analysis examined the relationship between self-efficacy, perceived social support, perceived organizational support, compassion satisfaction, burnout, secondary traumatic stress, and resilience (Table 3). Cook's distance revealed the data of nineteen samples with multivariate outliers. Thus, these samples were removed and conducted multiple regression analysis followed by path analysis with the remaining 288 samples (i.e., 93.81% of the collected data) after assessing the various assumptions (Table 4 & *Note* of Table 5) to test objectives 2 through 5 as outlined in the Aim and Objectives section. The rationale for this approach is that path analysis is a more advanced technique than standard regression, as it allows for the simultaneous examination of multiple direct and indirect relationships among variables [71]. Unlike regression, which primarily focuses on estimating direct effects on a single outcome variable, path analysis offers a more comprehensive understanding by modeling interconnected pathways and capturing indirect associations [71]. Additionally, path analysis was performed to assess the strength and direction of the relationships of the five explanatory variables (Fig 1) with resilience. However, the variable

secondary traumatic stress was excluded from the model because of its non-significant linear relationship with resilience. The goodness-of-fit of the model was evaluated using Maximum Likelihood estimation, along with a range of model fit indices (Table 5). While the assumptions of multivariate normality, homoscedasticity, and absence of autocorrelation were met, the presence of univariate non-normality raised concerns regarding the accuracy of standard errors, confidence intervals, and significance tests in the multiple regression analysis. In response to these concerns, the research team applied a bias-corrected 95% bootstrap confidence interval, using 1,000 resamples, to assess the significance of direct, indirect, and total effects among the variables, as recommended [72].

Considering the importance of comprehensive reporting as recommended by Kline [73], the researchers reported key model fit statistics, including $\chi^2$ with Normed Chi-Square ($\chi^2$: $df$ ratio), Root Mean Square Error of Approximation [$RMSEA$], Comparative Fit Index [$CFI$], and Standardized Root Mean Square Residual [$SRMR$]. These statistics were provided to offer a thorough evaluation of model fit in path analysis, ensuring that the model's adequacy and precision were adequately assessed. The standardized estimates of direct, indirect, and total effects of the explanatory variables (i.e., self-efficacy, perceived social support, perceived organizational support, compassion satisfaction, and burnout) on the outcome variable, i.e., resilience, were assessed from the final model of resilience (Table 6). The relationship between explanatory and outcome variables was deemed significant if the 95% confidence interval did not include zero, indicating a p-value of ≤.05.

### Ethics approval and consent to participate

Ethical approval was obtained from Institutional Review Committee of Patan Academy of Health Sciences, Lagankhel, Lalitpur (Ref: nrs2303071705: 7 March 2023). Formal permission was obtained from the authority of the nursing administration of Tertiary Hospital A. The rights of the respondents were protected by taking written informed consent using a written informed consent form and keeping the collected information confidential. One copy of the informed consent form was provided to each respondent for their future use. Anonymity was maintained by requesting respondents not to write their names on the questionnaires, keeping their informed consent forms from the questionnaire separate, and having the research assistants collect the data. Additionally, privacy was maintained by giving questionnaires separately to each respondent. In addition, the RAs collected the filled-out questionnaires individually. Social ethics was also applied by wearing masks, using sanitizer, maintaining physical distancing, and suggesting that the respondents and the RAs wear masks when meeting to prevent COVID-19 transmission risk and to ensure no harm to the respondents.

## Results

### Socio-demographic and job-related characteristics

Table 1 indicates that the nearly equivalent distribution of the respondents belonged to the age groups ≤30 years and >30 years, demonstrating an average age of 32.63 years ($SD$ = 7.14). Furthermore, the highest proportion of respondents were married (73.9%), held a bachelor's degree of education in nursing (75.6%), were employed as staff nurses (93.2%), had temporary/contract-based appointments (56.7%), and had >5 years of experience in the current hospital (56.7%). Additionally, there was a nearly equivalent distribution among respondents working in medical/surgical departments (30.6%) and those in emergency/critical care departments (29.6%).

### Descriptive analysis of study variables

Table 2 shows that all these variables exhibited absolute skew values <2 and absolute kurtosis values <7. Additionally, the highest percentage of respondents demonstrated a high level of self-efficacy (48.2%); intermediate level [i.e., neither low nor high] of resilience (51.8%); moderate level of perceived social support (75.2%), perceived organizational support (69.7%), compassion satisfaction (49.2%), burnout (53.1%), and secondary traumatic stress (45.0%). Descriptive

**Table 1. Respondents' socio-demographic and job-related characteristics.** $N = 307$.

| Variables | Frequency | Percentage (%) |
|---|---|---|
| Age in years ($X = 32.63$, $SD \pm 7.14$) | | |
| ≤ 30 | 151 | 49.2 |
| >30 | 156 | 50.8 |
| Marital Status | | |
| Married | 227 | 73.9 |
| Unmarried | 80 | 26.1 |
| Level of Education | | |
| PCL Nursing | 61 | 19.8 |
| Bachelor's Level in Nursing | 232 | 75.6 |
| Master in Nursing | 14 | 4.6 |
| Current Designation | | |
| Sister In-charge | 21 | 6.8 |
| Staff Nurse | 286 | 93.2 |
| Type of Appointment | | |
| Temporary/Contract | 174 | 56.7 |
| Permanent | 133 | 43.3 |
| Experience in Current Hospital (in years) | | |
| ≤ 5 Years | 133 | 43.3 |
| >5 Years | 174 | 56.7 |
| Currently Working Ward | | |
| Medical/Surgical Department | 94 | 30.6 |
| Emergency/Critical Care Department | 91 | 29.6 |
| Gynecology/Obstetric Department | 62 | 20.3 |
| Operation Theatre | 33 | 10.7 |
| Others | 27 | 8.8 |

*Note.* Others: Pediatric/Psychiatric Department.

analyses, including the mean, standard deviation, skewness, and kurtosis of the items assessing resilience, are presented in S1 Table; self-efficacy in S2 Table; perceived social support in S3 Table; perceived organizational support in S4 Table; compassion satisfaction in S5 Table; burnout in S6 Table; and secondary traumatic stress in S7 Table.

## Correlations between explanatory variables and resilience

Table 3 reveals the statistically significant relationships between the various factors (i.e., self-efficacy, perceived social support, perceived organizational support, compassion satisfaction, and burnout) and resilience. Notably, there was no significant relationship between secondary traumatic stress and resilience. Importantly, the analysis confirmed no issue of multicollinearity among the variables ($r < .90$). The correlations among the explanatory variables ranged from -.01 to.64. Furthermore, the curve estimation for linearity confirmed that all variables, except for secondary traumatic stress, were sufficiently linear for inclusion in the path analysis model of resilience.

## Model of and factors explaining resilience among nurses

Before conducting the path analysis, the research team performed multiple linear regression analysis to explore direct relationships among variables (Table 4).

**Table 2. Descriptive analysis of the respondents' resilience, self-efficacy, perceived social support, perceived organizational support, compassion satisfaction, burnout, and secondary traumatic stress N = 307.**

| Variables | Obtained Scores | | | | | | | | | |
|---|---|---|---|---|---|---|---|---|---|---|
| | Range | M | SD | Mdn | Q1 | Q3 | IQR | Skewness/SE of Skewness | Kurtosis/ SE of Kurtosis | Level (%) |
| Resilience [a] | 8-40 | 30.09 | 5.13 | 30 | 27.00 | 34.00 | 7.00 | -.54/.14 | .69/.28 | Low (28.0) |
| | | | | | | | | | | Intermediate (51.8) |
| | | | | | | | | | | High (20.2) |
| Self-efficacy [b] | 21-40 | 33.26 | 4.04 | 33 | 30.00 | 36.00 | 6.00 | -.33/.14 | -.18/.28 | Low (42.7) |
| | | | | | | | | | | Medium (9.1) |
| | | | | | | | | | | High (48.2) |
| Perceived Social Support [c] | 18-84 | 65.31 | 12.05 | 68 | 61.00 | 72.00 | 11.00 | -1.31/.14 | 2.08/.28 | Low (12.7) |
| | | | | | | | | | | Moderate (75.2) |
| | | | | | | | | | | High (12.1) |
| Perceived Organizational Support [d] | 0-48 | 27.45 | 10.38 | 27 | 23.00 | 34.00 | 11.00 | -.23/.14 | .19/.28 | Low (13.7) |
| | | | | | | | | | | Moderate (69.7) |
| | | | | | | | | | | High (16.6) |
| Compassion Satisfaction t-Score [e] | 22.96-62.85 | 50.00 | 10.00 | 51.77 | 42.90 | 58.42 | 15.52 | -.65/.14 | -.25/.28 | Low (20.8) |
| | | | | | | | | | | Moderate (49.2) |
| | | | | | | | | | | High (30.0) |
| Burnout t-Score [f] | 27.81-74.28 | 50.00 | 10.00 | 49.11 | 43.30 | 56.86 | 13.56 | .21/.14 | -.57/.28 | Low (24.1) |
| | | | | | | | | | | Moderate (53.1) |
| | | | | | | | | | | High (22.8) |
| Secondary Traumatic Stress t-Score [g] | 42.19-54.73 | 50.00 | 10.00 | 48.46 | 42.19 | 54.73 | 12.54 | .74/.14 | .35/.28 | Low (26.4) |
| | | | | | | | | | | Moderate (45.0) |
| | | | | | | | | | | High (28.7) |

*Note.* The level was categorized based on prior references: [a]: **Resilience** [51]: Low (Quartile First = 27), Intermediate (2nd and 3rd Quartile = 28 to 34, and High (4th Quartile = > 34). [b] **Self-efficacy** [53]: Low (< 33, i.e., below the median score), Medium (At the level of median score), and High (> 33, i.e., greater than the median score). [c] **Perceived Social Support** [59]: Low (<Mean − 1 SD, i.e., < 53.26), Moderate (Between Mean ± 1 SD, i.e., 53.26 to 77.36); and High (> Mean + 1 SD, i.e., >77.36). [d] **Perceived Organizational Support** [62]: Low (1 standard deviation below the mean), Moderate (Between -1 and +1 standard deviation), and High (1 standard deviation above the mean. [e] **Compassion Satisfaction** [67]: Low = Below 42.90 (Below 25th Percentile), Moderate = 42.90 to 58.42 (25th to 75th Percentile), and High = > 58.42 (Above 75th Percentile). [f] **Burnout** [67]: Low = Below 43.30 (Below 25th Percentile), Moderate = 43.30 to 56.86 (25th to 75th Percentile), and High = > 56.86 (Above 75th Percentile). [g] **Secondary Traumatic Stress** [67]: Low = Below 42.19 (Below 25th Percentile), Moderate = 42.19 to 54.73 (25th to 75th Percentile), and High = > 54.73 (Above 75th Percentile). Raw Total Scores of CS, BO, & STS were converted into z-score and then converted into t-score using these equations [i.e., tCS = (zCS × 10) + 50; tBO = (zBO × 10) + 50; and tSTS = (zSTS × 10) + 50].

Table 4 showed that Self-efficacy ($\beta = 0.38$, $p < .001$) and compassion satisfaction ($\beta = 0.19$, $p = .002$) had statistically significant positive associations with resilience. However, perceived social support showed a small, marginally significant association with resilience ($\beta = 0.10$, $p = .067$), while perceived organizational support and burnout were not significantly associated statistically significant association with resilience. Additionally, the tolerance and Variance Inflation Factor [*VIF*] statistics indicated the absence of multicollinearity among the variables (Table 4).

The hypothesized model comprised five exogenous and two endogenous variables (Fig 1). However, the analysis indicated that one of the exogenous variables (i.e., STS) did not meet the assumption of a linear relationship with resilience (Table 3). Consequently, STS was excluded from the final hypothesized model. Subsequently, the model fit analysis, comprising four exogenous and two endogenous variables, indicated that the hypothesized model demonstrated a good fit with the empirical data ($\chi^2 = .829$ [$p = .661$]), $\chi^2$: $df = .415$, $GFI = .999$, $AGFI = .990$, $RMSEA = .000$ (.000,.090], $PCLOSE = .810$, $SRMR = .010$, $CFI = 1.000$, $NFI = .998$, $NNFI = 1.026$ (Table 5).

**Table 3. Correlations between explanatory variables and resilience.** *N* = 307.

| Variables | 1 | 2 | 3 | 4 | 5 | 6 | 7 |
|---|---|---|---|---|---|---|---|
| 1. Self-efficacy | 1.00 | | | | | | |
| 2. Perceived Social Support | .32*** | 1.00 | | | | | |
| 3. Perceived Organizational Support | .22*** | .28*** | 1.00 | | | | |
| 4. Compassion Satisfaction | .37*** | .32*** | .32*** | 1.00 | | | |
| 5. Burnout | -.29*** | -.44*** | -.33*** | -.55*** | 1.00 | | |
| 6. Secondary Traumatic Stress | -.01 | -.23*** | -.15* | -.10 | .64*** | 1.00 | |
| 7. Resilience | .46*** | .29*** | .20*** | .36*** | -.29*** | -.06 | 1.00 |

*Note.* *: $p < .05$. **: $p < .01$. ***: $p = <.001$.

**Table 4. Factors associated with resilience applying multiple linear regression analysis.** *N* = 288.

| Variables | B | Standard Error | β | p-value | 95% CI [LL, UL] | Tolerance | VIF |
|---|---|---|---|---|---|---|---|
| Self-efficacy | .46 | .06 | .38 | <.001*** | [.33,.58] | .83 | 1.21 |
| Perceived Social Support | .04 | .02 | .10 | .067 | [-.003,.09] | .78 | 1.29 |
| Perceived Organizational Support | .02 | .02 | .04 | .443 | [-.03,.07] | .85 | 1.17 |
| Compassion Satisfaction | .09 | .03 | .19 | .002 | [.07,.32] | .65 | 1.53 |
| Burnout | -.01 | .03 | -.02 | .785 | [-.13,.10] | .65 | 1.54 |

*Note.* Dependent Variable: Resilience. * = *p* value significant at < 0.05 level (2-tailed). *** = *p* value significance at < 0.001 level (2-tailed). *B* = Unstandardized Beta, *β* = Standardized Beta, Adjusted $R^2$ = .295.

Likewise, with bias-corrected 95% bootstrap *CI*, the model showed that self-efficacy (*β* = .38, *p* = .003, *CI* = [.28,.48]), perceived social support (*β* = .10, *p* = .035, *CI* = [.01,.20]), and compassion satisfaction (*β* = .19, *p* = .002, *CI* = [.07,.31]) statistically significantly and positively influenced resilience. However, perceived organizational support (*β* = .04, *p* = .430, *CI* = [-.06,.14]) did not exhibit a statistically significant direct positive effect on resilience (Table 6 and Fig 3). Additionally, burnout (*β* = -.02, *p* = .762, *CI* = [-.15,.09]) did not demonstrate a statistically significant direct negative effect on resilience.

Likewise, perceived social support (*β* = .07, *p* = .003, *CI* = [.03,.12]) and compassion satisfaction (*β* = .12, *p* = .001, *CI* = [.08,.19]) had a statistically significant indirect and positive effect on resilience through self-efficacy. Moreover, the total effects of the pathways from perceived social support (*β* = .17, *p* = .005, *CI* = [.06,.28]) and compassion satisfaction (*β* = .31, *p* = .002, *CI* = [.19,.43]) to resilience were statistically significant (Table 6 & Fig 3). As a result, the combined effects of self-efficacy, compassion satisfaction, perceived social support, perceived organizational support, and burnout explained 31.0% of the variance observed in resilience ($R^2$ = .31). A post-hoc analysis was done to calculate the 95% confidence interval (*CI*), statistical power, and effect size using Dr. Daniel Soper's Free Statistical Calculators [75] with an Adjusted $R^2$ = .31, n = 288, and five explanatory variables. The findings were *CI* = [0.24, 0.38], an effect size = 0.45, and a statistical power = 1.0 at a 5% significance level (*α* = 0.05). The high statistical power indicates a minimal risk of committing a Type II error, thus providing strong confidence in the model's ability to identify significant relationships. Moreover, both perceived social support and compassion satisfaction demonstrated significant positive direct effects on self-efficacy, accounting for 17.0% of the variance observed in self-efficacy ($R^2$ = .17) (Table 6 & Fig 3).

## Discussion

This study adopted the cross-sectional path-analytical approach to construct and validate a model of resilience among 288 randomly chosen nurses in Nepal amidst the COVID-19 pandemic. The discussion centers on resilience levels

**Table 5. Model fit statistics of the hypothesized model. N = 288.**

| Statistical Test | Hypothesized Model | Reference Criteria of Goodness of Fit Value | Remarks on the Final Model |
|---|---|---|---|
| Absolute Fit Indices | | | |
| Chi-Square | .829 (p = .661) | p > .05 [74] | Good Fit |
| Normed Chi-Square ($x^2$: df ratio) | .829: 2 (.415) | 2:1 [74] | Good Fit |
| GFI | .999 | > 0.90 [74] | Good Fit |
| AGFI | .990 | ≥ 0.90 [74] | Good Fit |
| SRMR | .010 | ≤.08 [74] | Good Fit |
| RMSEA [95% CI], PCLOSE | .000[.000,.090],.810 | ≤.06 [74] | Good Fit |
| Incremental Fit Indices | | | |
| CFI | 1.000 | >.90 [74] | Good Fit |
| NFI | .998 | >.90 [74] | Good Fit |
| NNFI [TLI] | 1.026 | >.90 [74] | Good Fit |

*Note.* **(A) Achieved assumptions of regression analysis:** (1) no missing data; (2) no multivariate outliers: based on standardized residual (i.e., -2.36 to 2.72); (3) multivariate normality revealed through histogram and standardized residual and unstandardized residual with Kolmogorov Smirnov test (p = .200), and multivariate kurtosis C.R. value = 1.83; (4) homoscedasticity based on scatter plot of the standardized residuals; (4) bivariate and multivariate linearity based on scatter plots and Normal P-P Plot of Regression Standardized Residual. In addition, the linearity of relationships between exogenous (i.e., compassion satisfaction, burnout, perceived social support, and perceived organizational support) and endogenous variables (i.e., self-efficacy and resilience) was revealed through curve estimation for linearity methods except secondary traumatic stress variables. (5) non-multicollinearity based on (a) Pearson product-moment correlation except secondary traumatic stress [r = .20 to -.55] (see Table 3), (b) tolerance value [.65 to.85], and (c) VIF [1.21 to 1.54], and (6) autocorrelation (i.e., Durbin Watson value = 1.73). **(B) Achieved assumptions of path analysis:** Level of measurement of variables (i.e., interval level of measurement), recursive model (i.e., unidirectional relationship). Used reliable and valid instruments for reducing measurement errors. Employed meta-theory of resilience and resiliency, and findings of previous studies, and an over-identified model for reducing model specification error. **(C) Model Fit Statistics.** *GFI*: Goodness of Fit Index. *AGFI*: Adjusted Goodness of fit index. *SRMR*: Standardized Root Mean Square Residual. *RMSEA*: Root Mean Square Error of Approximation. *CFI*: Comparative Fit Index. *NFI*: Normed Fit Index. *NNFI*: Non-Normed Fit Index. *TLI*: Tucker-Lewis Fit Index.

**Table 6. Direct, indirect, and total effects of the model of resilience. N = 288.**

| Objective No. | Paths | Direct Effects | | | Indirect Effects | | | Total Effects | | | |
|---|---|---|---|---|---|---|---|---|---|---|---|
| | | β | p | 95% bias-corrected bootstrap CI | β | p | 95% bias-corrected bootstrap CI | β | p | 95% bias-corrected bootstrap CI | Decision |
| O-3 | CS→RES | .19 | .002** | [.07,.31] | | | | | | | Significant |
| | CS→SEF | .32 | .001** | [.22,.41] | | | | | | | Significant |
| O-5 | CS→SEF→RES | | | | .12 | .001** | [.08,.19] | .31 | .002** | [.19,.43] | Significant |
| O-3 | POS→RES | .04 | .430 | [-.06,.14] | | | | | | | Non-Significant |
| O-3 | PSS→RES | .10 | .035* | [.01,.20] | | | | | | | Significant |
| | PSS→SEF | .18 | .005** | [.07,.28] | | | | | | | Significant |
| O-5 | PSS→SEF→RES | | | | .07 | .003** | [.03,.12] | .17 | .005** | [.06,.28] | Significant |
| O-3 | SEF→RES | .38 | .003** | [.28,.48] | | | | | | | Significant |
| O-4 | BO→RES | -.02 | .762 | [-.15,.09] | | | | | | | Non-Significant |

*Note.* *: p ≤ .05. **: p < .01. BO: Burnout. CS: Compassion Satisfaction. PSS: Perceived Social Support. POS: Perceived Organizational Support. RES = Resilience. SEF: Self-efficacy. CI: Confidence Interval.

and their interplay with the variables. A comparative analysis with prior studies has been done to explore similarities and disparities, elucidating how these outcomes align with or diverge from existing research, especially considering the unique socio-cultural contexts in Nepal. The present study reports that the highest percentage of respondents had an

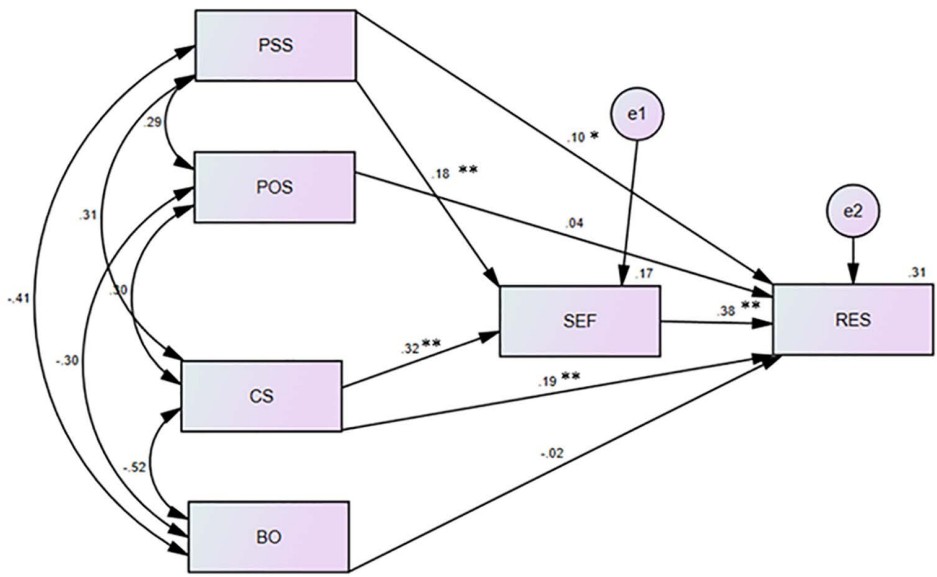

*N = 288*

Note. BO: Burnout. CS: Compassion Satisfaction. MPSS: Multidimensional Perceived Social Support. POS: Perceived Organizational Support. RES = Resilience. SEF: Self-efficacy. Excluded the path between secondary traumatic stress to resilience because of failing to meet the assumption of linearity. *: *p* <.05. **: *p* <.01. (χ2 = .829 [*p* =.661]), χ2: *df* = .415, GFI = .999, AGFI = .990, RMSEA = .000 (.000, .090], PCLOSE = .810, SRMR = .010, CFI = 1.000, NFI = .998, NNFI = 1.026.

**Fig 3. Model of resilience among Nepalese hospital nurses experiencing the COVID-19 pandemic.**

intermediate level of resilience, a finding consistent with cross-sectional studies of nurses conducted during the COVID-19 pandemic in the USA [52], Iran [76], Saudi Arabia [77], Lebanon [78], and China [79]. This consistency suggests that nurses globally may share similar resilience levels influenced by common pandemic experiences, collective coping mechanisms, and working environments. When employees exhibit a realistic understanding of their decision-making abilities and limitations, see change as an opportunity for growth, and maintain strong, supportive relationships with colleagues, they are better equipped to navigate challenges and setbacks [80]. Furthermore, personal goal-setting, a strong sense of humor, patience, and a high tolerance for adversity further foster their ability to maintain productivity and morale [80]. Additionally, the work efficiency and productivity of resilient employees might significantly increase and contribute positively to organizational success by adapting swiftly to changing circumstances and maintaining a proactive and optimistic outlook [80]. The evidence in the current and prior studies highlights that nurses might have some capacity to handle stress, adjust to challenges, and recover from hardship.

On the other hand, the contrasting findings in the cross-sectional studies showed that nurses in Turkey had high levels [81], whereas nurses in Palestine had low levels [82] of resilience. Moreover, a scoping review highlighted that most nurses experienced negative resilience because of fear of insufficient personal protective equipment and being infected by self-/family/colleagues [83]. This led to increased stress, depression, and anxiety [83]. Similarly, Nepalese health workers (i.e., nurses and doctors) during the COVID-19 pandemic had a low level of resilience [84]. A qualitative study during the COVID-19 pandemic recommended that adequate preparation, supportive working environments, motivation, individual and collective strategy enhancement, timely rewards, empowerment, counseling, and training are important in helping nurses cope with adversity and improve resilience [85]. The differences in resilience levels among studies may stem from

variations in the instruments used, study settings, the organizational climate, cultural contexts, educational backgrounds, geographical locations, and individual beliefs, including how their respective managers handled the nurses during the COVID-19 pandemic.

Concerning the hypothesized model, the results suggest that the model fits reasonably with the observed data, despite some non-significant paths. Nonetheless, the researcher retained the non-significant paths in the final model to preserve valuable information, as recommended by Grace-Martin [86]. In addition, Naqvi [87] emphasized that non-significant paths can provide future researchers with insights into determining sample sizes and addressing construct variations. Furthermore, Allen [88] argued that retaining non-significant variables helps ensure unbiased regression estimates and prevents invalid conclusions. However, there is limited evidence to compare this model with findings from previous studies. This study offers preliminary insights into the hypothesized relationships, contributing to the literature by presenting a comprehensive framework highlighting the personal, social, and organizational factors associated with resilience.

The current study reveals a statistically significant direct and positive effect of compassion satisfaction on resilience among nurses experiencing the COVID-19 pandemic. This finding aligns with previous research, including a mixed-method systematic review of 37 studies involving healthcare professionals (i.e., nurses and physicians) working at intensive care units [89]. Similarly, cross-sectional studies among nurses in Australia [90], Saudi Arabia [91], and mental health psychology practitioners in the UK [92] have reported comparable results. The consistent findings across various studies, including the current one, suggest a robust relationship between compassion satisfaction and resilience among healthcare professionals, particularly nurses facing the challenges of the COVID-19 pandemic. Compassion satisfaction, rooted in the fulfillment of helping others, aligns with key resilience-building elements like purpose and meaning in work. During the pandemic's stress, finding fulfillment in caring roles may provide vital emotional strength and help nurses cope effectively. Furthermore, the present study shows indirect and positive effects of compassion satisfaction on resilience through self-efficacy. The prior studies among Chinese clinical nurses revealed a statistically significant positive relationship between compassion satisfaction and self-efficacy [35]. Additionally, the Self-Determination Theory highlights that compassion satisfaction, by fulfilling intrinsic psychological needs for autonomy, competence, and relatedness, directly enhances self-efficacy, which in turn indirectly strengthens resilience, as individuals with higher self-efficacy are more likely to effectively cope with challenges [93].

Similarly, the current study reveals a statistically significant direct and positive effect of self-efficacy on resilience among nurses experiencing the COVID-19 pandemic, aligning with previous cross-sectional studies conducted among nurses in China two years after the COVID-19 pandemic [94] and mental health psychology practitioners in the UK [92]. Additionally, an integrative review of 17 articles on nurses' resilience during the pandemic [37] supported that self-efficacy contributed to resilience among nurses during the COVID-19 pandemic. In support of this, Schwarzer and Warner [95] highlighted that self-efficacy represents individuals' confidence to face life's challenges. Likewise, self-efficacy helps nurses adapt effectively and enables them to overcome adversity [96,97]. Lu et al. [98] further noted that nurses with high self-efficacy are better equipped to confront challenges and are less troubled by potential adverse outcomes because of their confidence in their ability to manage difficult situations. This confidence is reinforced by internal resources such as experience, knowledge, technical skills, and emotional insight that are cultivated over time [99].

Likewise, the present study reveals a statistically significant direct effect of perceived social support on resilience among nurses experiencing the COVID-19 pandemic. This finding is consistent with cross-sectional studies conducted among nurses in Iran [100] and China [60,101], as well as with evidence from diverse populations reported in a systematic review by Galanis et al. [102]. Moreover, an integrative review emphasized the significant contribution of family and community support in enhancing nurses' resilience, underlining the complex dynamics of resilience during the pandemic [37]. Furthermore, the present study reveals statistically significant indirect and positive effects of perceived social support on resilience through self-efficacy. This finding aligns with Hobfoll's conservation of resources theory [36], which posits that self-efficacy mediates the relationship between social support and resilience.

However, the current study shows a statistically non-significant effect of perceived organizational support on resilience among nurses experiencing the COVID-19 pandemic. In contrast, cross-sectional studies among nurses in Turkey [81] and China [65], as well as among healthcare workers in Hong Kong, Nepal, Vietnam, and Taiwan [84], reported a statistically significant positive relationship between perceived organizational support and resilience. Nevertheless, an integrative review among nurses showed that workplace/organizational support in terms of resources, policy, counseling, incentives, and appreciation contributed to resilience among nurses during the COVID-19 pandemic [37]. Similarly, the present study reveals a statistically non-significant effect of burnout on resilience among nurses during the COVID-19 pandemic. This finding stands in stark contrast to previous cross-sectional studies conducted in Canada [103], Australia [90], Iran [25], Saudi Arabia [91], and the UK [92], all of which consistently identified a significant negative association between burnout and resilience.

Conversely, the current study refutes the direct effect of perceived organizational support and burnout on resilience among nurses facing adversity. Prior research highlighted that toxic leadership [104]; lack of resources, increasing workloads, task complexity, reduced support, managerial challenges, workplace insecurity [105]; and workplace environment [106] significantly impacted nurses' psychological well-being, job satisfaction, and resilience. Similarly, Nepalese nurses face staffing shortages, workplace violence, inadequate nurse-to-patient ratios, frequent shift changes, and physically demanding duties, in addition to low pay, lack of recognition, and ethical dilemmas due to limited resources [39]. Likewise, they worked with insufficient equipment for patient care [40], excessive workloads [39,40], and multitasking pressure [40]. The lack of managerial support and guidance, long work hours and overtime without adequate rest and additional pay [40], and workplace hazards [39,40] further strain nurses' work environments. Additionally, a lack of clear performance evaluation criteria, limited promotion opportunities, difficulties when changing shifts or finding substitutes for leave, and strained relationships with supervisors exacerbate nurses' challenges [40]. Thapaliya et al. [41] further emphasized these factors, categorizing them into supply-side, demand-side, and maternal health service-related themes.

These environmental stressors result in adverse health effects [40]. Consistent with our observation, nurses faced the ongoing challenges of this pandemic for approximately three and a half years, continuously balancing the dual demands of infection prevention and patient care. Moreover, external stressors, such as leadership, organizational climate, and others mentioned earlier, may have played a more significant role in fostering resilience in such a unique and high-pressure environment during the COVID-19 pandemic than the perceived organizational support and burnout examined in this study. Importantly, these factors are not fully captured by standard instruments measuring perceived organizational support and burnout and thus may have contributed to nurses' capacity to adapt and sustain resilience amid unprecedented challenges. Therefore, nurses may require additional resources or targeted interventions to enhance their adaptability and resilience during and after the COVID-19 pandemic.

## Strengths and limitations of the study

Evidence highlighted that existing literature on factors associated with resilience has been fragmented, and evidence on developing and testing a comprehensive resilience model is scarce. Thus, this study provides unique findings by examining the factors explaining resilience among nurses amidst the COVID-19 pandemic. The researchers strengthened the study's robustness by employing proportionate stratified random sampling, ensuring a sample size that effectively represented nurses from Tertiary Hospital A. Anonymity was rigorously maintained to encourage valid and reliable responses, making the findings likely generalizable to all frontline nurses at Tertiary Hospital A during the pandemic. Additionally, the study utilized standard, valid, and reliable instruments.

However, several limitations should be considered. Firstly, the cross-sectional design of the study limits the ability to establish cause-and-effect relationships between explanatory factors and resilience due to temporal ambiguity. In addition, this study design was unable to adequately address the dynamic and complex nature of resilience and its determinants. To address this, future research should adopt a longitudinal study design to track changes in resilience over time, from

pre-pandemic through different pandemic phases and post-pandemic, or use experimental approaches to better understand these dynamics. Secondly, although Harman's single-factor test indicated that the data were free from common method bias, the reliance on self-report measures (i.e., questionnaires) may still introduce social desirability and response biases.

Moreover, this study was conducted in a single urban public hospital, which means the sample may not fully represent the characteristics of nurses in rural hospitals or private healthcare institutions in Nepal. Consequently, this could limit the generalizability of the findings, given the differences in the availability and accessibility of resources at Tertiary Hospital A compared to those in rural and private settings during the pandemic. Therefore, future research should aim to include diverse study settings, encompassing both rural and urban areas as well as public and private healthcare institutions. Furthermore, future research should incorporate in-depth qualitative exploration of the resilience journey across the pre-, during-, and post-pandemic phases. Notably, prior studies have suggested that leadership styles, workplace policies, cultural factors, optimism, self-compassion, stress, and coping may all influence nurses' resilience. Hence, future studies should employ a comprehensive resilience model using structural equation modeling to better capture the dynamic nature of resilience and explore additional factors contributing to resilience, particularly in the context of the COVID-19 pandemic.

Furthermore, although the 10-item Resilience Scale, 10-item General Self-Efficacy Scale, 8-item Perceived Organizational Support Scale, 12-item Multidimensional Scale of Perceived Social Support, and the 30-item Professional Quality of Life Scale (comprising three 10-item subscales for compassion satisfaction, burnout, and secondary traumatic stress) have been psychometrically validated in various languages, their Nepali versions remain insufficiently examined. While previous and current studies in Nepal have primarily assessed the reliability of these translated instruments, their validity, particularly among nurses, has not been thoroughly evaluated. Therefore, future research is warranted to validate these instruments within the Nepalese nursing context.

## Conclusion

The highest percentage of nurses exhibited intermediate levels of resilience, with the model showing a good fit with the empirical data. Statistically significant positive and direct relationships were found between compassion satisfaction, perceived social support, and self-efficacy with resilience. Additionally, compassion satisfaction and perceived social support had an indirect positive effect on resilience, mediated by self-efficacy. However, perceived organizational support and burnout did not demonstrate statistically significant direct relationships with resilience. Thus, this study recommends that healthcare organizations focus on enhancing compassion satisfaction, perceived social support, and self-efficacy to strengthen nurses' resilience while continuing to address organizational support and burnout as part of broader well-being initiatives.

### Implications of this study

This study adds to the existing literature by examining the relationship between self-efficacy, compassion satisfaction, social support, and resilience among nurses during the COVID-19 pandemic, an area that has not been thoroughly examined in prior research. The findings can assist hospital and nursing administrations in prioritizing training programs, resilience-building workshops, or creating policies and practices that support nurses in nurturing compassion satisfaction, perceived social support, and self-efficacy, ultimately fostering a more resilient workforce. By focusing on resilience-building initiatives and allocating resources to these efforts, healthcare systems can improve patient care quality, safety, and the well-being of their workforce.

### Highlights of the study

- Nurses had an intermediate level of resilience.

- The proposed model of resilience was well-aligned with empirical data.

- Self-efficacy, perceived social support [PSS], and compassion satisfaction [CS] had a direct and positive relationship with resilience.

PSS and CS positively influenced resilience through self-efficacy with its total effects.

## Supporting information

**S1 Table. Mean, standard deviation, skewness, and kurtosis of each item of the resilience.**
(DOCX)

**S2 Table. Mean, standard deviation, skewness, and kurtosis of each item of the self-efficacy.**
(DOCX)

**S3 Table. Mean, standard deviation, skewness, and kurtosis of each item of the perceived social support.**
(DOCX)

**S4 Table. Mean, standard deviation, skewness, and kurtosis of each item of the perceived organizational support.**
(DOCX)

**S5 Table. Mean, standard deviation, skewness, and kurtosis of each item of compassion satisfaction.**
(DOCX)

**S6 Table. Mean, standard deviation, skewness, and kurtosis of each item of burnout.**
(DOCX)

**S7 Table. Mean, standard deviation, skewness, and kurtosis of each item of secondary traumatic stress.**
(DOCX)

## Acknowledgments

The research team would like to thank all the nurses whose untiring efforts and cooperation made this study possible. We appreciate the experts who supported the cross-cultural content validation of the selected instruments. Finally, we express our sincere gratitude to all the administrative authorities of the study sites for their unwavering support. We are grateful to Ms. Judith Hall from English Language Partners in New Zealand for her editorial assistance.

## Author contributions

**Conceptualization:** Sarala KC, Rekha Timalsina, Shanta Dangol Shrestha, Praneed Songwathana.
**Data curation:** Sarala KC, Rekha Timalsina, Shanta Dangol Shrestha, Praneed Songwathana.
**Formal analysis:** Sarala KC, Rekha Timalsina, Shanta Dangol Shrestha, Praneed Songwathana.
**Funding acquisition:** Sarala KC, Rekha Timalsina, Shanta Dangol Shrestha, Praneed Songwathana.
**Investigation:** Sarala KC, Rekha Timalsina, Shanta Dangol Shrestha, Praneed Songwathana.
**Methodology:** Sarala KC, Rekha Timalsina, Shanta Dangol Shrestha, Praneed Songwathana.
**Project administration:** Sarala KC, Rekha Timalsina, Shanta Dangol Shrestha, Praneed Songwathana.
**Resources:** Rekha Timalsina.
**Software:** Rekha Timalsina.
**Supervision:** Sarala KC, Rekha Timalsina, Shanta Dangol Shrestha, Praneed Songwathana.

**Validation:** Sarala KC, Rekha Timalsina, Shanta Dangol Shrestha, Praneed Songwathana.

**Visualization:** Sarala KC, Rekha Timalsina, Shanta Dangol Shrestha, Praneed Songwathana.

**Writing – original draft:** Sarala KC, Rekha Timalsina, Shanta Dangol Shrestha, Praneed Songwathana.

**Writing – review & editing:** Sarala KC, Rekha Timalsina, Shanta Dangol Shrestha, Praneed Songwathana.

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
