## [Decision Letter · Decision Letter 0]

15 Jul 2025

PMEN-D-25-00140

Factors Explaining Resilience Among Nepalese Nurses of Tertiary-level Hospital Experiencing COVID-19 Pandemic: A Cross-sectional Study

PLOS Mental Health

Dear Dr. Timalsina,

Thank you for submitting your manuscript to PLOS Mental Health. After careful consideration, we feel that it has merit but does not fully meet PLOS Mental Health’s publication criteria as it currently stands. Therefore, we invite you to submit a revised version of the manuscript that addresses the points raised during the review process.

We look forward to receiving your revised manuscript.

Kind regards,

João Silvestre Silva-Junior, MD MSc PhD

Academic Editor

PLOS Mental Health

Journal Requirements:

1. We noticed you have some minor occurrence of overlapping text with the following previous publication(s), which needs to be addressed:

- https://doi.org/10.1186/s12912-018-0298-7

- https://doi.org/10.1016/j.ijdrr.2021.102756

- https://doi.org/10.1371/journal.pone.0284796

In your revision ensure you cite all your sources (including your own works), and quote or rephrase any duplicated text outside the methods section. Further consideration is dependent on these concerns being addressed.

Reviewers' comments:

Reviewer #1: 

The article is appropriate for the publication and is knowledgable.

a) Relaibility Analysis is missing.

b) Add details of scales used, regarding the author/s and year of publication.

c) Table of regression analysis shouold be added.

d) The format of the draft should be added as per APA format of manuscript.

e) Add description of Table 5.

Reviewer #2: 

The manuscript entitled "Factors Explaining Resilience Among Nepalese Nurses of Tertiary-level Hospital Experiencing COVID-19 Pandemic: A Cross-sectional Study" addresses a highly relevant and timely topic. The selection of the study population—Nepalese nurses working during the COVID-19 pandemic—is appropriate and fills a critical gap in the literature on resilience in low- and middle-income countries. The use of a theoretical model and path analysis adds methodological rigor and analytical depth to the study.

The following points are suggested to strengthen the manuscript:

1. Objectives in the Introduction:

Please ensure that the aims/objectives of the study are clearly stated at the end of the introduction, not only in the Methods section. This is important for aligning with PLOS editorial structure and improving clarity for readers.

2. Length of the Introduction:

The introduction is too long and includes extensive conceptual content that might be more effectively relocated to the discussion section. A more concise introduction focused on the study rationale, context, and knowledge gap is recommended.

3. Clarity on Instrument Validity:

It is unclear whether the psychometric tools used (e.g., CD-RISC, General Self-Efficacy Scale, Burnout Measure) were validated in the Nepali language or underwent formal cross-cultural adaptation. Please clarify this in the Methods section for each instrument and provide supporting references.

4. Ethics Section Placement:

The paragraph discussing ethical approval and informed consent is currently placed in the middle of the Methods section. It would improve the manuscript's structure to move this paragraph to the end of the Methods section, immediately after the description of statistical analyses.

5. Wording in the Conclusion:

The final sentence of the manuscript states that “perceived organizational support, burnout and resilience were not associated.” However, these variables were not analyzed jointly. To avoid misinterpretation, we suggest rephrasing to: "Perceived organizational support or burnout were not associated with resilience," which more accurately reflects the analysis performed.

---

## [Decision Letter · Decision Letter 1]

2 Oct 2025

Factors Explaining Resilience Among Nepalese Nurses of Tertiary-level Hospital Experiencing COVID-19 Pandemic: A Cross-sectional Study

PMEN-D-25-00140R1

Dear Dr. Timalsina,

We are pleased to inform you that your manuscript 'Factors Explaining Resilience Among Nepalese Nurses of Tertiary-level Hospital Experiencing COVID-19 Pandemic: A Cross-sectional Study' has been provisionally accepted for publication in PLOS Mental Health.

Best regards,

João Silvestre Silva-Junior, MD MSc PhD

Academic Editor

PLOS Mental Health